# Toward More Comprehensive Homologous Recombination Deficiency Assays in Ovarian Cancer, Part 1: Technical Considerations

**DOI:** 10.3390/cancers14051132

**Published:** 2022-02-23

**Authors:** Stanislas Quesada, Michel Fabbro, Jérôme Solassol

**Affiliations:** 1Medical Oncology Department, Institut Régional du Cancer de Montpellier (ICM), 34298 Montpellier, France; michel.fabbro@icm.unicancer.fr; 2Faculty of Medicine, University of Montpellier, 34090 Montpellier, France; j-solassol@chu-montpellier.fr; 3Montpellier Research Cancer Institute (IRCM), Institut National de la Santé et de la Recherche Médicale (INSERM) U1194, University of Montpellier, 34298 Montpellier, France; 4Department of Pathology and Onco-Biology, Centre Hospitalier Universitaire (CHU) Montpellier, 34295 Montpellier, France

**Keywords:** high-grade serous ovarian cancer, homologous recombination deficiency, BRCA, genomic scars, HRD assays, PARP inhibitors

## Abstract

**Simple Summary:**

High-grade serous ovarian cancer (HGSOC) is the most frequent and lethal form of ovarian cancer and is associated with homologous recombination deficiency (HRD) in 50% of cases. This specific alteration is associated with sensitivity to PARP inhibitors (PARPis). Despite vast prognostic improvements due to PARPis, current molecular assays assessing HRD status suffer from several limitations, and there is an urgent need for a more accurate evaluation. In these companion reviews (Part 1: Technical considerations; Part 2: Medical perspectives), we develop an integrative review to provide physicians and researchers involved in HGSOC management with a holistic perspective, from translational research to clinical applications.

**Abstract:**

High-grade serous ovarian cancer (HGSOC), the most frequent and lethal form of ovarian cancer, exhibits homologous recombination deficiency (HRD) in 50% of cases. In addition to mutations in *BRCA1* and *BRCA2*, which are the best known thus far, defects can also be caused by diverse alterations to homologous recombination-related genes or epigenetic patterns. HRD leads to genomic instability (genomic scars) and is associated with PARP inhibitor (PARPi) sensitivity. HRD is currently assessed through *BRCA1/2* analysis, which produces a genomic instability score (GIS). However, despite substantial clinical achievements, FDA-approved companion diagnostics (CDx) based on GISs have important limitations. Indeed, despite the use of GIS in clinical practice, the relevance of such assays remains controversial. Although international guidelines include companion diagnostics as part of HGSOC frontline management, they also underscore the need for more powerful and alternative approaches for assessing patient eligibility to PARP inhibitors. In these companion reviews, we review and present evidence to date regarding HRD definitions, achievements and limitations in HGSOC. Part 1 is dedicated to technical considerations and proposed perspectives that could lead to a more comprehensive and dynamic assessment of HR, while Part 2 provides a more integrated approach for clinicians.

## 1. Introduction

High-grade serous ovarian cancer (HGSOC) is the most frequent and lethal form of epithelial ovarian cancer (EOC) [1]. Despite substantial improvement in the clinical management of HGSOC, the all-stage 5-year overall survival (OS) rate remains at approximately 40% [2,3,4]. Thus, a better understanding of this disease is urgently required, from molecular deciphering to new therapeutic molecules. As such, a novel class of molecules, called polyadenosine diphosphate-ribose polymerase (PARP) inhibitors, emerged during the last decade. PARP inhibitors (PARPis) are based on homologous recombination deficiency (HRD), a molecular alteration that affects approximately 50% of HGSOC cases. In parallel with PARPi development, HRD assays have been developed to provide clinicians with accurate estimates of homologous recombination status. As such, PARPis, coupled with HRD assays, led to substantial improvements in HGSOC prognosis [5].

However, HRD assays remain controversial, notably due to their technical and medical relevance [6,7]. By bridging the gap between molecular and clinical considerations, these companion reviews will present the evidence to date regarding HRD definitions, achievements and limitations in EOC, with the aim of providing physicians and researchers involved in HGSOC management with a holistic perspective, from translational research to clinics. Part 1 focuses on molecular and technical considerations, describing: 1. the main components of HRD through a dichotomic approach (i.e., causes and consequences); 2. the rationale, development and technical performance of current HRD assays; and 3. the limitations inherent to current HRD assays and the axes of research and proposed perspectives that could lead to a more comprehensive and dynamic assessment of HRD, with the aim of improving its predictive value. The companion paper (Part 2) focuses on clinical considerations and, notably, the impact of PARPis in the clinic.

## 2. From Ovarian Cancer Genetics to Homologous Recombination Defects

### 2.1. HGSOC Predispositions: Germline Alterations and Affected Pathways

Discoveries in observational studies and family studies led to the determination of genetic factors involved in EOC risk, notably those associated with the so-called hereditary breast and ovarian cancer (HBOC) syndrome caused by germline mutations in *BRCA1* or *BRCA2* (*BRCA1/2*). Based on The Cancer Genome Atlas (TCGA) database, it is estimated that 15–20% of HGSOC patients carry a germline mutation in *BRCA1/2* genes, leading to a cumulative risk of developing EOC by the age of 80 years of 44% and 17%, respectively, versus 1.4% in the general population [8,9]. *BRCA1* mutation carriers exhibit OC at a younger age than *BRCA2* mutation carriers. The high prevalence of *BRCA1/2* mutations led to the formulation of international guidelines to integrate genetic testing, or at least genetic counseling, upon EOC diagnosis, particularly in the context of a familial history of OC [10,11,12,13]. Furthermore, prophylactic risk-reducing bilateral salpingo-oophorectomy has been shown to be an effective prevention strategy in germline *BRCA1/2* carriers [14]. It also should be noted that *BRCA2* germline mutation increases the risk of prostate and pancreatic cancers [8,13]. A few cases of constitutive epimutations (i.e., aberrant hypermethylation) of the *BRCA1* promoter have been reported [15,16]. Other germline mutations have also been described, such as *RAD51C*, *RAD51D*, *PALB2*, *BARD1* or *BRIP1*, representing a cumulative frequency of approximately 5% [17,18]. Notably, inherited mutations in mismatch repair genes (mainly *hMLH1*, *hMSH2*, *hMSH6* and *PMS2*), which lead to Lynch syndrome, are also known to increase the risk of OC, although they mainly exhibit endometrioid histology [19]. Beyond germline alterations, EOC depends on a few crucial pathways. Indeed, *TP53* (which encodes the tumor suppressor p53) is mutated in approximately 96% of HGSOC cases and is considered an early event and driver mutation of cancer progression [20,21]. Data from TCGA have shown recurrent mutations in a restricted set of genes: *BRCA1*, *BRCA2*, *NF1*, *RB1* and *CDK12* [9]. Furthermore, HGSOC is also characterized by frequent chromosomal instability through DNA gain/loss, leading to tumor suppressor gene (TSG) loss and oncogene amplification [22].

Strikingly, several genes altered in HGSOC are involved in DNA repair through the homologous recombination (HR) process. Briefly, single-strand breaks (SSB) are processed through the base excision repair (BER) mechanism, mainly involving PARP proteins (of whom PARP1 is the most characterized). When facing double-strand break (DSB), cells will use different repair mechanisms: homologous recombination (HR), non-homologous end joining (NHEJ) and microhomology-mediated end joining (MMEJ). While the HR process leads to faithful DNA repair by using a homologous template, end resects are directly ligated by NHEJ, potentially leading to small insertions/deletions (indels). MMEJ also leads to specific indels, which are longer than the ones occurring during NHEJ [23].

### 2.2. BRCA1 and BRCA2: Key Players in EOC

At the genetic level, alterations to *BRCA1* and *BRCA2* are the most frequent and best -characterized alterations in the HR pathway. Indeed, both germline and somatic deleterious mutations in *BRCA1/2* genes (referred to as *gBRCA** and *sBRCA**, respectively) have been shown to promote HGSOC [9,24]. *BRCA1/2* play key roles in genome integrity maintenance and their alteration are an early event in the EOC carcinogenetic process; indeed, the loss of the first allele of *BRCA1* (or *BRCA2*) is a facilitating event for *TP53* loss, both of them leading to EOC development [25]. The most frequent and well-known alterations are short mutations in *BRCA1/2* genes, leading to coding sequence disruption (through missense, nonsense or frameshift mutations) that subsequently inactivate proteins or result in dominant-negative mutations [26]. According to the two-hit Knudson hypothesis, in the context of *gBRCA**, all the cells in the patient already carry an inactive copy of *BRCA* (the first hit); thus, the loss of the second allele (the second hit), which mainly occurs through loss of heterozygosity (LOH), is the only step needed to produce HRD [27]. In contrast, patients without *gBRCA** need two hits to develop HRD. *gBRCA1** and *gBRCA2** are estimated to occur in 8% and 6% of HGSOC cases, respectively. Furthermore, *sBRCA1** and *sBRCA2** are found in an additional 4% and 3% of cases, respectively [9,28]. Notably, it has been suggested that *BRCA1* haploinsufficiency due to *gBRCA1** could be sufficient to initiate tumorigenesis through the *TP53* mutation, without needing a second hit [29]. This hypothesis is supported by the fact that retention of the normal allele (i.e., absence of LOH) has been reported to occur in 7% and 16% of HGSOC cases that carry *gBRCA1** or *gBRCA2**, respectively [30]. Nevertheless, in the context of *sBRCA**, LOH is considered to be near-universal; additionally, a few studies have reported that short mutations could be an alternative second hit [31,32].

Recently, multimegabase large rearrangements (LRs) at the *BRCA1/2* loci have been suggested to lead to HRD, accounting for approximately 16% of patients [33]. Owing to their different structures, the BRCA1 and BRCA2 proteins exhibit distinct functions in the cell. Aside from HR, in which the role of BRCA2 remains unclear, BRCA1 exhibits a variety of functions in the cell, such as cell cycle regulation, chromatin remodeling, replication fork protection and apoptosis [34,35].

In addition to direct alterations to DNA sequences, epigenetic mechanisms can also lead to loss of expression of *BRCA1*. Indeed, *BRCA1* promoter methylation (-*CpG+*) has been characterized for decades [36]. Data from TCGA revealed that the *BRCA1-CpG+* mechanism leads to HRD in approximately 11% of HGSOC cases [9]. According to a meta-analysis based on 16 studies, *BRCA1-CpG+* is found in approximately 16% of EOCs [37]. A recent study, which analyzed 88 EOCs, found *BRCA1-CpG+* and *BRCA2-CpG+* in 19.3% and 4.6% of the cases, respectively [38]. This latter finding could imply that *BRCA2-CpG+* is a cause of HRD, although it has been classically considered a quite rare event. Interestingly, *BRCA* mutations and *BRCA1-CpG+* are almost mutually exclusive [9].

### 2.3. Beyond BRCA: The BRCAness Concept

Although initially described with *BRCA1/2**, HRD was later characterized in the context of wild-type *BRCA* (*BRCA^wt^*); therefore, the so-called “BRCAness” phenotype, encompassing any HRD not caused by a direct *BRCA* alteration, was identified [39]. In *BRCA^wt^*, biallelic mutations in HR genes that lead to BRCAness have been described. They represent approximately 5% of cases and mainly concern mutations to *RAD51C*, *RAD51D*, *BRIP1* and *PALB2* [40,41,42]. In addition to point mutations, LRs that affect genes other than *BRCA1/2*, such as *RAD50* and *NBS1*, have been reported [43]. Furthermore, HRD can be the consequence of EMSY (a BRCA2-interacting transcriptional repressor) amplification, an alteration found in approximately 5% of cases [44,45]. Notably, a specific subset of HGSOC cases exhibits *CCNE1* amplification, a molecular carcinogenetic pathway associated with HR proficiency and a poor prognosis [46].

In addition to BRCAness due to genetic mutations, promoter methylation in other HR genes, such as *RAD51C* and *PALB2*, has been described in HGSOC [26,47]. Apart from protein-coding RNAs, the role that microRNAs (miRNAs), typically studied in the regulation of gene expression through translation inhibition and mRNA degradation, play in the carcinogenetic process has begun to emerge, with specific miRNA signatures associated with OC [48,49]. Deregulation of some miRNAs, such as miR-509-3p and miR-211, through HR inhibition, has been implicated in platinum-sensitive (Pt-S) cancers [50,51]. Long noncoding RNAs (lncRNAs), which exhibit a wide range of physiological functions, such as transcription regulation and scaffolding within cells, have been linked to OC risk, carcinogenesis and prognosis [52,53]. For instance, *PCAT*-1 leads to HRD in prostate cancer by suppressing BRCA2 [54]. Although several lncRNAs have been shown to be involved in HR, their actual etiologic impact on HRD still needs to be assessed in EOC [55]. Interestingly, it has been suggested that miRNA and lncRNAs deregulations are caused by alterations in methylation, implying crosstalk between these two epigenetic processes.

Chromatin dynamics rely mainly on posttranslational modifications (PTMs), nucleosome positioning and spatial genome organization; these modifications are involved in complex crosstalk with DNA methylation [56]. Currently, there is no clinical proof that such epigenetic modifications can directly lead to HRD in the context of OC [57]. However, as chromatin can act as a gene silencer and a barrier to efficient DNA repair, it appears plausible that chromatin remodeling could partly explain (or at least participate in) HRD in OC [58]. Indeed, several histone PTMs have been described in the context of DSBs, notably participating in the selection between the HR and NHEJ pathways [59,60]. Interestingly, H2AX phosphorylation at serine 139 (the so-called γ-H2AX) is considered the hallmark of DSBs [61]. H4K16 acetylation and H3K36 trimethylation act synergistically and are required for HR after a DSB; loss of these markers leads to inefficient HR [62,63]. Bromodomain-containing protein 9 (BRD9), which has been shown to be mutated in EOC, is essential for acetylation of RAD54 and its interaction with RAD54 is likewise necessary for efficient HR [64].

In conclusion, although not easily translatable to routine clinical practice, many of the intricate epigenetic mechanisms involved in HR and its deficiency have been described thus far.

### 2.4. Consequences of HRD

Owing to the democratization of next-generation sequencing (NGS), intensive oncological research has been conducted to investigate correlations between specific cancer types and distinct sets of DNA alterations (the “mutation signature”). In addition to histomolecular correlations, data on specific carcinogens (e.g., tobacco smoke) and tumor aggressiveness have led to molecular classifications [65,66]. Thus, the consequences of HRD can be classified into three different categories: DNA alterations, epigenetic markers and functional phenotypes (Figure 1).

#### 2.4.1. Genetics

DNA alterations can be divided into two classes: microlesions (affecting a single to a few nucleotides) and macrolesions (affecting larger DNA regions, up to megabases). At the microlesional scale, *BRCA* alterations (caused by *gBRCA1/2**, *sBRCA1/2** or *BRCA1-CpG+*) lead to a specific base substitution pattern, named “signature 3” [67,68,69]. In contrast to the other defined signatures, the *BRCA1/2**-associated signature exhibits a relatively equal distribution of different base substitutions (i.e., transitions and transversions) that are quite homogeneous across the gene. Furthermore, it is characterized by a high number of microhomology-mediated deletions, which are consequences of HRD and the compensatory use of the MMEJ mechanism. Interestingly, *BRCA1/2** leads to a seven-fold increase in base mutagenesis through HRD, thus promoting oncogenesis and intratumoral heterogeneity [70]. In addition to *BRCA* alterations, signature 3 has been observed in other HR-related genes, leading to a *BRCAness* molecular signature [71,72,73]. Notably, this signature is not present in the context of incomplete inactivation of HR-related genes [35].

At the (sub)chromosomal scale, HRD leads to gross rearrangements and aberrations [74]. Several types of LR have been described as consequences of HRD. Branded with the generic term “genomic scars”, they constitute a permanent fingerprint of HRD-related global genomic instability [75]. The literature tends to use “mutational signatures” and “genomic scars” to refer to microlesions and macrolesions, respectively; however, these are interchangeable. HRD leads to a specific panel of copy number alterations through deletions, duplications, inversions and translocations. Indeed, *BRCA1/2** breast tumors exhibit a specific genomic profile in array comparative genomic hybridization (aCGH), a profile that can also be found in tumors that exhibit *BRCAness* [76,77].

More precisely, three types of alterations are enriched in tumors with HRD: LOH, large-scale state transitions (LSTs) and telomere allelic imbalance (TAI). *Stricto sensu*, LOH is an allelic imbalance (i.e., loss of equilibrium between paternal and maternal alleles); it can be either “copy defective” (i.e., a simple loss of one of the alleles and subsequent haploidy in a given locus) or “copy neutral” (i.e., no loss of diploidy) [78]. In the context of HRD, LOH refers to deletions >15 megabases (Mb) but less than a whole chromosome [79]. TAI is defined as an allelic imbalance >11 Mb in subtelomeric regions [80]. Conversely, LST refers to an allelic imbalance >10 Mb in size between adjacent genomic regions, a phenomenon caused by translocations and insertions/deletions [81]. Notably, in the context of HRD, these alterations are found across the genome, and their global enrichment is reflected in the global genomic instability score (GIS) [82].

#### 2.4.2. Epigenetic Markers

In addition to genetic alterations, HRD is also associated with epigenetic alterations. However, the link between cause and consequence is harder to establish for epigenetic alterations. Indeed, the different HRD-related epigenetic alterations can precede, maintain or occur after an HRD phenotype develops.

The simplest layer of epigenetics is based on specific gene expression profiles (GEPs), secondary to DNA mutations [83]. Most of the studies assessing epigenetics to date have been performed in the context of *BRCA**. Seminal studies have shown specific GEPs are associated with *BRCA** tumors, with a distinct panel depending on the gene affected (*BRCA1* versus BRCA2**) and on the etiology (germinal *versus* sporadic); intriguingly, a partial overlap exists among these and a subset of *BRCA^wt^* HGSOC cases, suggesting a common HRD-associated GEP [84,85,86]. A 60-gene-specific panel associated with *BRCAness* has been described [87]. Specifically, HRD exhibits a distinct core GEP [88]. Notably, in the context of *gBRCA1**, even healthy fallopian tissue exhibits a specific GEP, suggesting an epigenome-modifying influence [89,90]. The partial divergence observed between *BRCA1** and *BRCA2** tumors probably relies on the pleiotropic function of BRCA1, which is not restricted to DSB management [91,92].

Several epigenetic markers have been associated with HRD. Importantly, BRCA1 has been shown to negatively regulate Polycomb-repressive complex 2, a major chromatin remodeling enzyme involved in stem cell-state maintenance through transcriptional repression of histone H3K27 trimethylation [93,94]. Following HRD, tumors exhibit specific PTMs, with lower levels of H4K12/16-acetylation and overexpression of histone deacetylase 6 [95,96]. In patients with *gBRCA1/2**, apparently “normal” fallopian tissue still carries a reprogrammed epigenome with a specific methylome [97]. The inefficiency of HR leads to a lack of RAD51 recruitment at DSB sites (the so-called “RAD51 foci”) [98,99]. Furthermore, a subnetwork of 30 co-expressed proteins distinguishes HRD *versus* non-HRD HGSOC [96].

#### 2.4.3. Functional Consequences

At the functional level, HRD tumors are metabolically distinct; indeed, they rely more on oxidative phosphorylation than glycolysis [100]. Owing to error-prone DSB processing, HRD tumors have a higher tumor mutational burden (TMB), neoantigen load and HLA-I expression [101]. This is in accordance with the increase in tumor-infiltrating CD8+ lymphocytes and the higher expression of PD-1/PD-L1 proteins observed in *BRCA1/2** HGSOC, reflecting an “immunologically hot” phenotype [102].

Clinically, HRD tumors tend to be more sensitive to platinum-based regimens and PARPis, although differences exist depending on the underlying HRD mechanism [98,103]. Platinum-based treatments mainly rely on DSB generation via crosslinks with DNA, a phenomenon that is highly toxic to cells in the context of HRD [104]. In terms of PARPis, the sensitivity of HRD tumors is based on their synthetic lethality (SL). This concept relies on the fact that cancer cells harbor gene defects that are not lethal per se, but that become lethal when combined with a defect in another gene [105]. When using PARPis, SL occurs because of the inability of HRD tumors to manage DSBs. Some PARPs (a large family of 17 proteins that participates in several cellular pathways through the ADP-ribose PTM and whose deregulation is implicated in carcinogenesis) are involved in DNA repair. PARP1, which is the most characterized PARP, plays a key role in SSB repair mainly through BER, although its role in MMEJ has recently been described [106,107]. Consequently, PARPis impairs the BER pathway, leading to SSB accumulation and progression to DSBs. In HR-proficient cells, these DSBs will be processed, allowing continued cell viability. In contrast, HRD cells accumulate DSBs, ultimately leading to apoptosis. Interestingly, recent studies have shown that PARPis have other roles, such as stalling the replication fork, stalling or trapping PARP1 on DNA (leading to protein-DNA adducts) and subsequent cell death in HRD cells [108]. In the context of HGSOC, seminal studies showed both in vivo and in vitro a specific SL occurring with PARPis in a *BRCA*-deficient context [109,110]. Subsequently, this breakthrough class of agents started to emerge in randomized clinical trials with substantial improvements in patients with HRD HGSOC, leading to approval of three molecules to date: olaparib, niraparib and rucaparib. Clinical considerations will be developed in Part 2 of this review. Noteworthy, HRD HGSOC spontaneously tends to have improved progression-free survival (PFS) and OS, which is in part due to better treatment responses [111].

Consequently, an accurate evaluation of the HRD status of cases remains essential, for both prognosis and theranostics.

## 3. HRD Companion Assays in Clinical Practice

Owing to the potent impact of HRD status on HGSOC management, several assays have been developed for HRD evaluation; while some have been used in research thus far, others are currently used as “companion diagnostic” (CDx) assays prior to PARPi prescription. This section will focus on clinically validated and/or routinely used tests (Table 1), while tests currently under investigation will be discussed in the fourth part of this review. This review will only focus on technical considerations (performances and limitations of each test), while the clinical considerations (i.e., the relevance of evaluating HRD status as a biomarker for PARPi prescription and response) will be detailed in the related paper. To date, three CDx assays have currently received FDA approval for OC [112]. HRD evaluation mainly relies on two strategies (following the cause *versus* consequences dichotomy): searching for mutations in HR-related genes (mainly *BRCA**; the causes of HRD) and/or the presence of “genomic scars” (the consequences of HRD).

### 3.1. BRCA Mutations

Currently, the mutation statuses of *BRCA1* and *BRCA2* can be either evaluated through gene-specific (i.e., *BRCA1/2* targeted sequencing) or multipanel testing, the latter detecting potently targetable non-BRCA alterations. Based on central sequencing confirmation, the *gBRCA* tests provided by different companies have been shown to produce reliable results concerning molecular alterations, with a concordance of >95% [113]. The rate of variants of unknown significance (VUS), which was originally 84% during initial *BRCA* testing development, has dropped to approximately 10% due to VUS reclassification and refinement, notably through the Consortium of Investigators of Modifiers of BRCA (CIMBA) [114]. However, non-*BRCA* HR-related genes, when used in some randomized controlled trials (RCTs) evaluating PARPis or Pt-S in HGSOC, have produced conflicting results and are currently under investigation [11,12,40,42].

Depending on the context, *BRCA* analysis can be performed either in blood samples (i.e., constitutive) or directly within the tumor (*tBRCA*). While detection of *gBRCA** in blood samples generally implies altered *BRCA* within the tumor, the presence of a mutation within the tumor (i.e., *tBRCA**) from formalin-fixed paraffin-embedded tissue (FFPE) can be due either to *sBRCA** or *gBRCA**. Current international guidelines and practices recommend *tBRCA** testing, at a minimum, for newly diagnosed HGSOC, though systematic *gBRCA* evaluation recommendations vary between guidelines [10,11,22,115,116].

As *gBRCA** represents the vast majority of *tBRCA** and has important implications for the hereditary risk of breast and ovarian cancers (HBOC) and screening of patient relatives, current guidelines systematically recommend referral to genetic counseling upon *tBRCA** detection [11,12,115,116]. Evaluation of *gBRCA* needs to be performed both at the micro- and macro-lesion scales, as LR constitutes a non-negligible fraction of *BRCA* alterations. Following the decision of the USA Supreme Court concerning the ineligibility of *BRCA* gene sequencing patents, many institutes and companies have developed their own assays [117].

MyriadGenetics (MG), which initially characterized the sequence of *BRCA*, developed the BRACAnalysis^®^ CDx in accordance with its extensive experience with BRACAnalysis^®^ of HBOC [118]. In 2014, BRACAnalysis^®^ CDx (BA-CDx) became the first FDA-approved CDx, along with olaparib, for treating patients with *gBRCA** advanced OC who previously received ≥3 lines of chemotherapy [119]. BA-CDx relies on sequencing genomic DNA obtained from whole blood samples collected in EDTA. The entire coding sequences of the *BRCA1/2* genes are included (as well as promoter regions and intron/exon boundaries). Point mutations and short indels are analyzed with polymerase chain reaction (PCR) and Sanger sequencing, while large rearrangements (LR-including deletions/duplications) are detected with multiplex PCR and a proprietary system entitled BART^®^ CDx.

There are five different classifications of results: positive for a deleterious mutation; genetic variant-suspected deleterious; genetic variant, favor polymorphism; genetic VUS; and no mutation detected. The VUS classification is the one of highest concern for geneticists and oncologists, with a moving interpretation according to scientific discoveries. MG uses its own private database, which relies on its extensive experience with *BRCA* analysis for determining hereditary cancer risk and variant classifications.

To MG (consistent with the American College of Medical Genetics and Genomics recommendations), mutations are considered deleterious if they fall in one of the following categories: nonsense and frameshift mutations occurring at/or before the last known deleterious amino acid position of the affected gene; deletions/duplications of entire exons; LR leading to frameshifts; and mutations/LR based on the data derived from the linkage analysis of high-risk families, functional assays, biochemical evidence, statistical evidence, and/or demonstration of abnormal mRNA transcript processing [120]. MG classification is performed by a committee of experts (i.e., board-certified laboratory and medical directors, research scientists, genetic counselors and variant support specialists) and further enriches their proprietary database [121].

According to the technical information given by MG, analytical validation studies (i.e., nonclinical studies) of BA-CDx were performed on a set of 110 samples containing single nucleotide variants (SNVs), deletions of up to 40 base pairs (bp) and insertions of up to 10 bp, with a validated NGS-based assay as comparator [121]. Agreement analyses included 100% positive percent agreement (PPA), negative percent agreement (NPA), and overall percent agreement (OPA). BART^®^ CDx accuracy was evaluated by using a validated microarray assay on a set of 103 patients (with 29 samples positive for a large rearrangement in *BRCA1* or *BRCA2*). BART^®^ CDx yielded valid results for 98 samples, with concordance between the 2 tests for 97 samples (the discordant sample was identified as duplication by BART^®^ CDx and triplication by the reference assay) [121].

Clinical validation was performed on samples from several studies that evaluated the effect of PARPis on OC. Notably, a subset of 61 samples from Study 42 (NCT01078662), initially tested locally, was retrospectively tested with BA-CDx, and provided a 96.7% (59/61) concordance rate (the 2 discordant samples included 1 sample without a callable result with BA-CDx and another with a different variant classification result). Patients from the SOLO1 RCT (NCT01844986) enrolled either through prospectively testing with BA-CDx (n = 181) or local testing (n = 210). Of these 210 patients, 208 were retrospectively tested, and concordance was achieved for 207 of them (98.5%) [113,121].

The main limitation of the BA-CDx assay is that it only detects *gBRCA**; as *gBRCA** represents approximately 70% of *BRCA*-mutated patients; a negative BA-CDx result does not rule out *sBRCA**. Other limitations of the BA-CDx include the lack of general detection of insertions that differ from duplications and unequal allele amplification (and subsequent false-negative results) from rare polymorphisms under primer sites [121]. BA-CDx is currently FDA-approved as a CDx assay in three distinct situations where *gBRCA** evaluation is required prior to PARPi prescription, as detailed in Table 1.

### 3.2. Genomic Scars

Two commercially available CDx assays have been prospectively validated for the evaluation of genomic scars thus far: FoundationFocus CDxBRCA-LOH^®^ (FF-CDx) from FoundationMedicine (FM) and MyChoice CDx^®^ (MC-CDx) from MG [11,22]. Both tests combine *tBRCA* sequencing and genomic scarring evaluation. As *BRCA* analysis is performed on tumoral tissue, it does not distinguish between *gBRCA** and *tBRCA*.*

FF-CDx is based on comprehensive (i.e., including characterization of point mutations and indels) deep NGS, and thus has a >95% sensitivity and >99% positive predictive value; the starting material is FFPE, either in block form or on at least 10 unstained slides, with a minimum of 20% malignant tissue for *BRCA1/2* analysis [122,123]. It relies on whole-genome shotgun library construction and hybridization-based capture, amplification and sequencing of the constructed library. Interestingly, while the capture process targets approximately 1.5 Mb of the human genome (including all coding exons of 310 cancer-related genes, introns or noncoding regions of 35 genes and >3500 SNVs located throughout the genome), the FF-CDx only reports results for *BRCA1/2*, raising the question of “lost data”. Subsequently, *tBRCA1/2** (including SNVs and indels up to 13 bp) are detected through a custom analysis pipeline, with a 5% mutation allele frequency (MAF) cutoff (lowered to 1% for SNVs and 3% for indels mutations in hotspots). *tBRCA** is considered deleterious when it leads to premature stop codons (PSCs) anywhere in *BRCA1/2* coding regions (with the exception of the *BRCA2* PSC at position K3326 and 3′ downstream), splice site alterations (defined as mutations at intron/exon junctions, ±2 bp from exon starts/ends) or deleterious missense alterations (according to the curated list based on the Breast Cancer Information Core database).

An analytical validation study of FF-CDx was performed on a set of 36 *tBRCA** (including SNVs, deletions up to 12 bp and insertions up to 4 bp) and 44 *tBRCA* wild-type (*tBRCA^wt^*) samples, with a validated NGS-based assay as a comparator. Agreement analyses were as follows: 100% PPA, 94.9% NPA and 97.3% OPA. The limits of detection (LODs, defined as the minimal allele frequency necessary to detect a given lesion) vary by type of alteration: 6% for SNVs and indels ≥12 bp in non-repetitive regions and 15.3% for deletions in homopolymer regions. Clinical validation was performed on a subset of the samples from Study 10 (NCT01482715) and the ARIEL2 (NCT01891344) study. The clinical bridging study, which compared FF-CDx versus local testing for *BRCA1/2* evaluation, was performed on 67 samples and showed 97% PPA, 100% NPA and 97.9% OPA [124]. LOH was calculated through an almost genome-wide (i.e., the 22 pairs of autosomes) analysis of the >3500 SNVs detected in the hybridization-based capture process, leading to a global score that reflected the percentage of genomic LOH. Tumors were defined as “LOH high” (≥16%) or “LOH low” (<16%), corresponding to HRD-positive (HRD+) and HRD-negative (HRD-) statuses, respectively [125]. Notably, a positive HRD result was produced if the tumor was “LOH high” and/or exhibited a *tBRCA**. Laboratory validation of the LOH component of the assay indicated the LOD was ≥35% DNA tumor content for a reliable analysis; furthermore, the accuracy of the LOH evaluation (calculated through inter-run reproducibility) was lower when the LOH value approached the 16% cutoff. Notably, “LOH high” initially had a different cutoff (≥14%), as it was defined through TCGA analysis for the ARIEL2 study [126].

Interestingly, the FF-CDx evolved in parallel with RCTs that evaluated the PARPi rucaparib. Indeed, FoundationMedicine initially proposed a test entitled FoundationFocus CDxBRCA© that only focused on *tBRCA* alterations (point mutations and short indels); this test received FDA approval in 2016, along with the PARPi rucaparib, for patients with *tBRCA**-associated advanced OC with ≥2 lines of chemotherapy [127]. This test subsequently included LOH analysis following the ARIEL2 and ARIEL3 (NCT01968213) studies that evaluated rucaparib, with the aim of determining the HRD status of patients with mutations beyond *tBRCA**; the LOH analysis was performed with FM T5 NGS, which is an assay developed for clinical trials and has the same characteristics as the marketed FF-CDx [126,128]. Interestingly, 6% (ARIEL2) and 8.7% (ARIEL3) of the LOH evaluations provided inconclusive results.

Importantly, FF-CDx is no longer available as a stand-alone assay but is provided within the more general FoundationOne CDx© (F1-CDx), driving a comprehensive multi cancer analysis. Indeed, beyond the *tBRCA** and LOH evaluations, F1-CDx can detect mutations in a panel of 324 genes captured through the hybridization-based process, including copy number alterations, TMB, MSI and specific gene rearrangements [22]. The results are provided in three classes: CDx claims, cancer mutations with evidence of clinical significance, and cancer mutations with potential clinical significance. Comparison with the LOH evaluation of FF-CDx produced 97.5% PPA, 95.1% NPA and 96.7% OPA.

According to its technical information, the F1-CDx assay allows LR identification; however, its concordance with other validated methods has not been evaluated. Consequently, confirmatory validation is required upon copy number alterations that affect *BRCA* (except for whole *BRCA1/2* homozygous deletion). Furthermore, although this test theoretically has an LOD of >20% tumor purity for LR (importantly, this value is given for all HR pathway genes), the clinical bridging study using the SOLO1 (which only enrolled patients with *tBRCA**) samples gave conflicting results. Indeed, 368 (94.1%) patients from SOLO1 were retrospectively tested, and 335 had a valid F1-CDx result. Of these 335 patients, a deleterious mutation in *BRCA1/2* was not confirmed in 22 cases. Twelve of these discrepancies were due to differences in variant classification (i.e., different criteria between F1-CDx and local testing assays); the remaining 10 patients actually had LR (≥1 exon deletions or duplications), indicating a substantial lack of sensitivity for LR detection from the F1-CDx assay; however, this resulted in FDA approval of F1-CDx as a CDx for this indication [129]. Recently, FoundationMedicine has marketed the FoundationOne Liquid^®^ CDx for detecting *tBRCA** (with a parallel analysis of the 324-gene panel) directly in a blood sample; however, this test does not detect LOH and has only been validated prior to rucaparib treatment [130]. It should be noted that multigene panel testing, although time- and cost-efficient due to its comprehensive content, also increases the risk of detecting VUS. Interestingly, although FF-CDx for *tBRCA** detection is a prerequisite for PARPi prescription for two distinct indications in OC, the third FDA-approved indication (HRD evaluation prior to rucaparib maintenance as a second-line treatment) is not biomarker-driven. Indeed, a positive HRD status is considered predictive of efficacy and to indicate enhanced PFS.

The development of MC-CDx, which is supported by MG’s expertise with BRACAnalysis^®^, has also advanced; its GIS calculation differs from that of FF-CDx. The correlation between LOH, LST and TAI with *BRCA1/2** and Pt-S has been previously described [80,81,82]; subsequently, the superiority of the correlation among these three measurements was shown in comparison to each individual component [131]. Consequently, the GIS (proprietary score of MG- GIS_MG_) consists of the unweighted numeric sum of LOH, LST and TAI. HRD positivity is currently defined by a GIS_MG_ ≥ 42 and/or *tBRCA1/2**. The GIS_MG_ ≥ 42 threshold was set following the analysis of a training cohort that consisted of 497 chemotherapy-naïve BC patients and 561 EOC patients whose *BRCA1/2* status was known. Thus, by evaluating HRD scores in this cohort, the GIS_MG_ positivity cutoff was predefined with a 95% sensitivity for detecting tumors with *BRCA1/2** or *BRCA-CpG+*; this test was then evaluated for its ability to identify Pt-S triple-negative BC (TNBC) in a neoadjuvant setting [132].

Following its initial validation in TNBC, MC-CDx was investigated in several RCTs, such as PRIMA, VELIA and PAOLA-1, which evaluated PARPis (niraparib, veliparib and olaparib, respectively) in EOC [133,134,135]. Importantly, similar to FF-CDx, the GIS positivity threshold of MC-CDx (GIS_MC_) has been set at different values. Indeed, while it was initially proposed at GIS_MG_ ≥ 42, another cutoff of GIS_MC_ ≥ 33 (corresponding to the first percentile of HRD scores observed in *tBRCA1/2** tumors) has also been used, such as in the VELIA trial, with the goal of avoiding false-negatives [42,46,134].

Similar to FF-CDx, MC-CDx relies on a multistep process based on hybridization-based capture and NGS (i.e., fragmentation, end repair and adenylation, adapter ligation, library construction/amplification, hybridization and capture, sequencing and data analysis). The hybridization process was performed through a custom Agilent SureSelect© capture array consisting of pangenomic probes at 54091 SNV sites and 685 probes for *BRCA1/2* exons and exon boundaries. Normalized base and exon coverage of *BRCA1/2* were calculated to detect LRs. MC-CDx exhibited a less sensitive performance in *BRCA1/2** detection (notably for LRs) than BA-CDx, as it is performed through NGS and on biopsies (implying tissue heterogeneity). Indeed, indels >25 bp were less frequently detected than whole gene duplications/deletions. The LODs were as follows: 7.2% for an SNV, 6.6% for a <10 bp deletion, 6.3% for a <10 bp insertion, 5.9% for a ≥10 bp deletion, 30% for ≥3 exons LR and 50% for 1–2 exons LR.

For BA-CDx, MC-CDx uses the MG proprietary classification score for variant classification. Comparison with a validated NGS-based assay on 209 FFPE clinical specimens from cancer patients (5 *tBRCA1/2** GIS_MC_ −; 71 *tBRCA1/2^WT^* GIS_MC_ −; 66 *tBRCA1/2** GIS_MC_ +; 61 *tBRCA1/2^WT^* GIS_MC_ +) indicated 99.9% PPA (95% lower limit confidence of 99.7%), 100% NPA for *BRCA1/2* SNVs/indels and 100% OPA for *tBRCA* LR. Concordance analysis gave high fidelity results, both for GIS status (98.5% OPA, 97.4% NPA and 98.1% OPA) and HRD status (98.5% OPA, 98.6% NPA and 98.5% OPA). In 136 FFPE samples harboring *tBRCA1/2^WT^*, a 0% false-positive rate was observed, and a MAF threshold of 5% was defined, as no spurious variants were observed above this cutoff. The MC-CDx results were not affected by necrosis of the tumor area up to 60%.

The SOLO1 sample (n = 391), already tested with BA-CDx (in whole blood) for *gBRCA**, retrospectively provided clinical validation of MC-CDx. FFPE DNA samples from 298 patients were used: 284 patients were confirmed to carry *tBCRA**, 8 were not, and 6 samples failed the test. Pathogenic LR, which was detected at the germline level by BA-CDx in 15 patients produced the following results within the tumors: detection in 12 (80%) cases, absence of detection in 1 (6.7%) case because of the known limit of MC-CDx detection and 2 (13.3%) unanalyzable samples because of the poor quality of the tumor specimens. In the almost “real-life” conditions of clinical trials, MC-CDx indicated an unknown HRD status in 10–20% of patients: 10% (46/463 patients-all causes) in QUADRA (NCT02354586), 12% (137/1140 patients-all causes) in VELIA (NCT02470585), 15% (54/350; 26 inconclusive results and 28 inadequate/missing specimens) in NOVA (NCT01847274), 20% (163/806; 70 inconclusive results and 93 inadequate/insufficient specimens) in PAOLA-1 (NCT02477644) and 15% (111/733; 80 inconclusive results and 31 inadequate/insufficient specimens) in PRIMA (NCT02655016). Notably, when focusing on trials that detailed the cause of the lack of HRD information (i.e., NOVA, PAOLA-1 and PRIMA) and by examining only the missing data caused by failed tests (i.e., inconclusive results in tested patients), it appears that MC-CDx fails to provide a valid result in 8.1–11.4% of tests.

As a consequence of this prospective validation, which was based on RCTs and will be detailed in the companion paper (Part 2), these two FDA-approved assays led to a better definition of therapeutic sensitivity (notably for PARPis) and have been included as part of the EOC management in the European Society for Medical Oncology (ESMO) and American Society of Clinical Oncology (ASCO) recommendations [11,115].

## 4. HRD Definitions: Current Technical Concerns and Perspectives

Despite yielding considerable progress in HGSOC management (which is detailed in the companion paper), validated HRD assays currently suffer from several technical limitations that can be schematically categorized as preanalytical, analytical or postanalytical. Furthermore, medical considerations, which will be developed in the companion paper, should also be assessed.

### 4.1. Technical Concerns

The technical concerns can be further divided into three categories: preanalytical, analytical and postanalytical. First, the different FDA-validated HRD assays (i.e., BA-CDx, MC-CDx and F1-CDx) do not exhibit the same performance on clinical samples and are therefore not interchangeable. When comparing MC-CDx (positive if ≥42%), the percentage of LOH (analogous to the measurement of HRD status through F1-CDx; positive if ≥16%) and an 11-gene panel (consisting of genes involved in the HR pathway; positive if a pathogenic variant is found), it appears that many HRD+ patients (according to MC-CDx) are not detected through the other methods. Indeed, up to 46% of HRD+ (MC-CDx) patients were missed by the percentage of LOH and the 11-gene panel. Moreover, using a lower cutoff (i.e., GIS ≥ 33%), such as used in the VELIA trial, resulted in missing up to 61% of patients [136].

#### 4.1.1. Preanalytical

According to the different RCTs, approximately 5–10% of samples are unfit for HRD processing. This can be the consequence of insufficient starting material, paucity of tumor cellularity, or poor quality or degraded specimens.

Another aspect that should be considered is sample heterogeneity, which occurs at two distinct levels. Typically, a processed sample exhibits < 100% tumor cellularity, meaning that the analysis will include a fraction of normal tissue that can interfere with HRD evaluation, either by lowering the GIS or preventing *tBRCA** detection; importantly, this limit does not exist for *gBRCA**, as all cells harbor the mutation. Moreover, each tumor exhibits different lineages and clonal evolutions, with possible discordant HRD statuses. In recent years, tumor heterogeneity has emerged as a cornerstone of treatment resistance, including resistance to platinum and PARPis. This intratumoral heterogeneity is also present between distinct tumors, for instance, between primary and secondary lesions [137].

#### 4.1.2. Analytical

BA-CDx, while being the gold standard for *gBRCA* analysis, detects only germinal mutations and consequently fails to detect *sBRCA**, which occurs in approximately 30% of *BRCA*-mutated HGSOC cases. Therefore, a negative result from the BA-CDx blood test does not rule out the presence of *sBRCA*.*

Setting aside BA-CDx, the tumoral HRD assays suffer from high LODs (explained in the companion review). While manufacturers indicate that a minimum value of 20% tumor cellularity is required for processing, at least 30–35% tumor cellularity is needed to evaluate GIS (or risk a false-negative). Furthermore, beyond the LODs related to tumor cellularity, HRD assays evaluating *tBRCA** barely detect LRs, although LRs are present in up to 16% of HGSOC patients, thus leading to false *tBRCA^wt^* statuses [33]. Recently, SIGNPOST showed that approximately 20% of *gBRCA** was missed by the initial *tBRCA** assessment, with the stunning revelation that none of the LRs of the cohort (representing 11% of *gBRCA**) were detected by the *tBRCA** evaluation [138].

#### 4.1.3. Postanalytical

Even if the starting material is sufficient for the requirements of the manufacturer, approximately 5–10% of processed samples still give inconclusive results regarding the HRD status. For *BRCA1/2* genes, the main concern is misclassification. Although recent years have led to an improved understanding of the functions and molecular alterations of BRCA1/2 (and consequently to more accurate classifications), discrepancies still exist within databases [139,140,141]. While the accumulation of data led to important refinements in the classification of *BRCA* mutations, and despite the synchronization of classification according to guidelines and data sharing, reclassification still frequently occurs [142]. In addition to clear pathogenic variants, “likely pathogenic” variants are frequently included in RCTs and in FDA/EMA approval, although the latter does not result in absolute proof of its pathogenic nature. Importantly, some variants are considered “pathogenic” based on a proven deleterious effect observed in a germline context. This “pathogenic” status is then extrapolated to *sBRCA**; however, *gBRCA** and *sBRCA** may have distinct effects at both the molecular and cellular levels, just as BRCA1 and BRCA2 do not share exactly the same functions. Furthermore, *tBRCA* analysis does not define zygosity; indeed, *tBRCA** can be homozygous and/or heterozygous in the sample, potentially leading to distinct effects. Orthogonal to clinical considerations, F1-CDx (which analyses over 300 genes unrelated to HR and other DNA alterations) also assesses “off-target” multigene testing, raising the risk of detecting VUS without increasing the possibility of detecting pathogenic variants (and subsequently proposing targeted therapy) [143].

At the GIS level, as discussed in the previous section and in the companion paper, defining the ideal threshold value is a matter of heated debate. Indeed, beyond providing the most accurate and precise HRD assay, the main risk of using an unfit threshold is misclassification. This falls into two categories: false-positives (FP) and false-negatives (FN). An FP represents an HRD-positive result when the sample was actually HRD-negative (or HR-proficient); an FN is when an HRD-positive sample is labeled HRD-negative. In MC-CDx, the test is positive if the GIS_MG_ is ≥42 (initially validated with TNBC and corresponding to a 95% sensitivity for detecting tumors with *BRCA* alterations); this test has been evaluated with different cutoffs. For instance, in the VELIA trial, a lower cutoff (GIS ≥ 33) was used, with the aim of preventing FNs. Moreover, within *BRCA^wt^* patients, HRD is not a predictive biomarker for PARPi sensitivity. Conversely, a retrospective study based on TCGA data and using a backward strategy (i.e., moving from GIS-based stratification toward clinical/molecular data) showed that using a threshold of GIS ≥ 63 led to accurate HGSOC classification among the subtypes, correlating well with prognosis and the presence of *BRCA1/2** [144]. Consequently, the “magic 42” still needs to be more precisely defined, with potential variation according to tumor type and clinical context.

Aside from these technical issues, medical considerations should also be taken into account, particularly the timing of HRD evaluation, to obtain a broader perspective. These considerations will be developed in Part 2 of this review.

### 4.2. Emerging Strategies for Accurate and Dynamic Assessment of HRD

Emerging strategies for HRD assessment mainly occur along three axes: molecular tools for HRD assessment, dynamic assays (i.e., functional assays) for evaluating HRD status, and more global strategies (including nomograms). One of the main considerations for refining HRD evaluation, beyond its relevance, is that a biomarker described in basic research should be feasible in clinical practice, taking technical, economic and temporal issues into account. For instance, metabolomics studies or spheroid cultures have shown that HRD tumors exhibit a specific profile; nevertheless, to date, these techniques appear unfit for routine clinical application [100,145].

Several strategies regarding molecular assays to determine HRD have been deployed, with various results. Notably, the primary aim is improved identification of patients who would benefit from frontline treatment with PARPis, as these drugs suffer the same limitations as existing CDx (i.e., genomic scars). Many private companies have developed their own tools, but they will not be discussed here, as those tools failed to show better results than those of MC-CDx. However, some interesting tools will be outlined. Signature 3, which is based on SNVs, correlates with HRD and platinum sensitivity [71]. A relevant tool, entitled Signature Multivariate Analysis (SigMA), allows HRD-associated mutational signatures to be detected directly from targeted gene panels [146]. HRDetect, which is based on whole-genome sequencing and the incorporation of six distinct HRD-related signatures that predict *BRCA1*/2 alterations, is a promising assay that provides almost 100% detection sensitivity [147]. Both signature 3 and HRDetect are “backward strategies”, meaning that they predict a *BRCA1/2* status from secondary molecular signatures. A recent paper showed that HRDetect outperforms %LOH (F1-CDx) and is equally efficient as GIS_MC_ for HRD identification [148]. Another emerging field is represented by the analysis of LRs; biomarkers, such as *CCNE1* and *ESMY* amplifications within LRs are associated with HR proficiency and deficiency, respectively [9,45,149]. Furthermore, specific profiles of LRs are associated with better prognosis and with Pt-S, implying potent sensitivity to PARPis [150,151]. Other promising candidates, such as *RAD50* deletion, *RB1* loss and *BRD4* amplification, could improve HRD detection [43,152,153].

While mutations in non-*BRCA* HRR-related genes provide conflicting results (only homozygous deletion in *PTEN* or *CHK1* is putatively associated with HRD), a more accurate evaluation could come from integrating clonal composition [144]. Indeed, by integrating NGS metrics (i.e., determining if variants correspond to clonal or subclonal mutations), it was recently shown that mutations in HRR-related genes were associated with OS and Pt-S only if they were clonal [154].

At the epigenetic level, *BRCA1* (and, to a lesser extent, *BRCA2)* methylation profiles should not be overlooked, although they are not routinely examined. The frequency of methylation is not anecdotal, as it has been reported in 19.3% (*BRCA1*) and 4.6% (*BRCA2*) of 92 *BRCA^wt^* cases, according to a retrospective study. Acquired loss of *RAD51C* promoter methylation is associated with PARPi resistance: more precisely, the presence of heterozygous methylation leads to resistance, while homozygous methylation leads to sensitivity [155]. Several studies have assessed HGSOC methylomes and revealed distinct methylation profiles linked to Pt-S or, conversely, primary/secondary platinum resistance (Pt-R); these could constitute putative biomarkers to evaluate in the context of PARPis [156,157,158].

Beyond deciphering specific alterations, such as DNA mutations or epigenetic markers, comprehensive approaches that collect and assemble common genomic, epigenomic and functional data would result in better molecular dissection and the development of new biomarkers [159,160]. Indeed, by developing multilayer (i.e., genome/exome, SVs, transcriptome, miRNome, proteome, methylome and proteome) and integrated molecular identity cards (“multiomics”), such as that performed by the TCGA but with a more HRD-oriented view, we could subsequently select “core biomarkers” with higher sensitivities and specificities, leading to a more accurate evaluation of HRD status [161,162,163,164]. As such, rather than extensive and expensive approaches that would be difficult to translate to clinical practice, we could define a subset of distinct biomarkers through distinct techniques that would increase the sensitivity/specificity of current CDx assays and decrease the number of inconclusive cases.

Sequencing ctDNA in blood samples, which has already been validated by the F1 L-CDx assay, allows direct evaluation of a *tBRCA1/2* status [165]. In the near future, an iterative analysis could provide clues as to primitive or secondary resistance, depending on the presence of reverse mutations, as it has been shown that they are associated with resistance to PARPis [166]. One method would be via direct assessment of GIS in blood samples. Although it is not currently performed on ctDNA, intriguing papers have shown that it is feasible to analyze SVs in blood [167,168,169]. The molecular signatures of miRNAs present in blood samples have been detected and showed an association with early OC and prognosis, thus demonstrating their feasibility as potent surrogate markers for HRD [170,171,172,173]. Furthermore, other serum biomarkers are under study and currently debated, but their translation into clinical practice seems difficult [174]. Moreover, ascitic cancer cells should be considered, as they provide a means to obtain material with less invasive processes than biopsies; these cells are easily obtained and well represent the mutations (including SVs), DNA methylation and intratumoral heterogeneity found in EOC [175]. Thus, “liquid biopsies” could represent an easy-to-use method for iterative sampling [176].

At the functional level, several techniques have been developed in the past decade with the aim of evaluating the actual HRD status [99]. Indeed, apart from investigating the causes (e.g., DNA mutations) and consequences (e.g., genomic scars) of HRD, functional assays directly monitor the process itself. Functional tests can potentially overcome the inherent limits of current assays, which require either a genetic alteration (i.e., affecting non-*BRCA* HR-related genes) or the calculation of GIS (which reflects a history of HRD). However, similar to current CDx assays, they do not predict the sensitivity of HRD-unrelated PARPis.

One of the techniques with a high potential for clinical applications is the REcombination CAPacity (RECAP) assay [177]. When a DSB occurs and is managed with HR, RAD51 (one of the downstream effectors of BRCA1/2) attaches to these sites to facilitate sister chromatid invasion. Thus, measuring RAD51 nuclear foci provides a direct evaluation of the efficiency of HR, irrespective of its etiology. In the context of HRD, these foci are absent.

HRD evaluation through RAD51 foci was initially developed in the laboratory and subsequently applied in different protocols, as ex vivo or through patient-derived xenograft assays, to predict patient sensitivity to platinum/PARPis and OS [178,179,180,181]. One of the main drawbacks of the RECAP assay is that it requires fresh tissue as the starting material as well as the induction of DNA damage, since the test relies on tissue irradiation and subsequent visualization of foci formation (with distinct cutoffs of foci corresponding to functional or deficient HR), limiting its application in clinics (as tumoral tissue is frequently processed as FFPE). However, as HRD tumor cells exhibit spontaneous important DSBs, an evolution of these assays has emerged. Though originally focused on BC samples, the assay was subsequently directly performed on FFPE samples, showing a correlation between RAD51 foci and HRD status [182,183]. Recently, an adaptation to ovarian and endometrial cancer FFPE tissues, entitled RAD51-FFPE, was developed [184]. Interestingly, RAD51-FFPE paved the way for clinical applications by optimizing the test via calibration of a threshold corresponding to functional HRD. This led to high sensitivity, as it allowed *BRCA*-deficient and HRD tumors to be detected in 90% and 87% of cases, respectively. Therefore, the next step will be its integration into clinical studies to evaluate its performance as a predictive biomarker.

## 5. Synthesis and Concluding Remarks

HGSOC, the most frequent and aggressive form of OC, represents an important challenge for researchers and clinicians. Half of these cases show HRD, which has specific causes and consequences. In terms of etiology, HRD is mainly caused by genetic and epigenetic alterations, with *BRCA1* and *BRCA2* best characterized thus far. In addition to *BRCA1/2*, many other lesions are involved in the etiology of HRD, leading to the development of the *BRCAness* phenotype. HRD has specific consequences at both molecular (e.g., genomic instability) and clinical (e.g., PARP inhibitor sensitivity) levels.

Based on its prevalence and its theranostics potency, HRD represents a major molecular factor that must be understood to improve HGSOC management. Three CDx assays currently have FDA approval for the identification of HRD status, helping physicians prescribe PARP inhibitors. However, there is an urgent need for novel assays, such as functional assays, to be developed and integrated into RCTs for clinical validation.

## Figures and Tables

**Figure 1 cancers-14-01132-f001:**
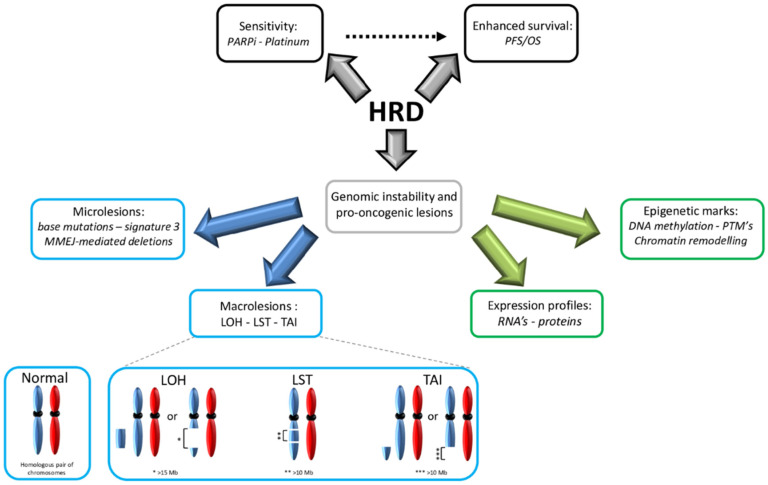
**Genomic instability as a consequence of HRD.** HRD shows genomic instability and pro-oncogenic lesions responsible for multiple genetic (blue squares) and epigenetic (green squares) events characterizing tumor progression and defining PARPi sensitivity and survival enhancement (black squares). Signature 3 refers to a specific base substitution pattern, which is the consequence of HRD. Abbreviations are as follows: HRD: homologous recombination deficiency; LOH: loss of heterozygosity; LST: large-scale transition; MMEJ: microhomology-mediated end-joining; OS: overall survival; PFS: progression-free survival; PARPi: poly (adenosine diphosphate-ribose) polymerase inhibitor; PTM: posttranslational modification; TAI: telomere allelic imbalance.

**Table 1 cancers-14-01132-t001:** Current FDA-approved CDx for HRD evaluation in ovarian cancer.

CDx Test	Sample Requirement	BRCA1/2 Analyses ^1^	BRCA1/2 Technical Concerns and Limits of Detection (LOD) ^1^	GIS Evaluation ^1^	GIS LOD ^1^	HRD + Cutoff ^1^	Cost (in $)	FDA Approvals
BRACAnalysis^®^ (MG)	● 5–7 mL (blood)	● gBRCA ● Targeting: - full coding sequence - flanking regions ● PCR + Sanger on AB^®^ ABI3730 (SNV, indels) ● Multiplex PCR with BART^®^ pipeline (LR)	● sBRCA* are not detected ● Less detected alterations: - LR - deletions > 40 bp - insertions > 10 bp	NA	NA	NA	0–295 ^2^	● 2014: OC with ≥3 L → OLA (gBRCA*) ● 2018: aEOC in P/CR → OLA (1Lm; gBRCA*) ● 2018: rEOC if ≥2 L → RUCA (gBRCA*)
MyChoice^®^ (MG)	● FFPE ● Block/≥10 slides ● ≥25 mm^2^ ● ≥20% tumor cells ● DNA: 30–200 ng	● tBRCA ● Hybridization-based capture + NGS ● Illumina^®^ HiSeq2500 ● Customized pipeline: - Indels - SNV - LR (exons + promoters)	● No distinction between sBRCA* and gBRCA* ● Less detected alterations: - indels > 25 bp - whole gene deletions ● Allele frequency LOD’s: - ≈7% (SNV) - ≈6% (<10 bp indels) - ≈30% (≥3 exons LR) - ≈50% (1–2 exons LR)	Pangenomic LST+LOH +TAI	≥30%	GIS ≥ 42 and/or tBRCA*	4040	● 2019: rEOC (≥3 L/Pt-s) → NIRA (HRD+) ● 2020: aEOC in C/PR → OLA + bevacizumab (1Lm; HRD+)
FoundationOne^®^ ^3^ (FM)	● FFPE ● Block/≥10 slides ● ≥25 mm^2^ ● ≥20% tumor cells ● DNA: 50–1000 ng	● tBRCA ● Hybridization-based capture + NGS ● Illumina^®^ HiSeq4000 ● Customized pipeline: - Indels - SNV - LR (coding exons) - Rearrangements	● No distinction between sBRCA* and gBRCA* ● Poorly detected alterations: - >13 bp indels - polyT regions - specific LR (≥1 exon indels, inversions, transversions, HD) ● Allele frequency LOD’s: - ≈6% (SNV, ≥12 bp indels in non-repetitive regions) - ≈15% (homopolymer regions) - ≈8% (BRCA2 HD) - ≈20% (rearrangements)	Pangenomic LOH	≥35%	LOH ≥ 16 and/or tBRCA*	5800	● 2016: rEOC with ≥2L ^4^ → RUCA (tBRCA*) ● 2018: aEOC in P/CR → OLA (1Lm; tBRCA*) ● 2018: rEOC in C/PR ^5^ → RUCA (2Lm)

Abbreviations are as follows: 1Lm = first-line maintenance; 2Lm = second-line maintenance; ≥2 L = 2 or more previous lines of chemotherapy; ≥3 L = 3 or more previous lines of chemotherapy; aEOC = advanced epithelial ovarian cancer (including primitive peritoneal and fallopian tube cancers); CDx = companion diagnostic; C/PR = complete/partial response to platinum-based chemotherapy; FFPE = formalin-fixed paraffin-embedded; FM = FoundationMedicine; gBRCA = germline BRCA; GIS = genomic instability score; HD: homozygous deletion; HRD (+) = homologous recombination deficiency (positive); LOH = loss of heterozygosity; MG = MyriadGenetics; NA = not applicable; NIRA: niraparib; OC = ovarian cancer; OLA: olaparib; Pt-s = platinum-sensitive; rEOC = recurrent epithelial ovarian cancer (including primitive peritoneal and fallopian tube cancers); RUCA: rucaparib; tBRCA = tumoral BRCA. Nota bene: ^1^ data is presented according to the technical information provided by the manufacturers and laboratory validation studies; complementary technical information, particularly for clinical validation studies, is provided within the text. ^2^ The cost of BRACAnalysis^®^ CDx is highly dependent on medical insurance coverage in the USA. ^3^ The F1-CDx also detects microlesions and macrolesions in 324 genes, selected gene rearrangements, and MSI and TMB. ^4^ In this context, FoundationOne Liquid CDx (performed on a whole blood sample) is also FDA-approved but does not provide LOH evaluation. ^5^ In this context, rucaparib is not biomarker-driven, but a positive HRD status is predictive of its efficacy and indicates improved progression-free survival.

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
