# Peer review of "Toward More Comprehensive Homologous Recombination Deficiency Assays in Ovarian Cancer, Part 1: Technical Considerations"

_cancers, 2022, doi:10.3390/cancers14051132_

Round 1
Reviewer 1 Report
This is a part of two related reviews. In this review, the authors focus on the molecular and technical aspects of homologous recombination deficiency (HRD) in the diagnosis and treatment of patients with High-grade serous ovarian cancer (HGSOC). They summarize the causes and consequences of HRD in ovarian cancers. They also discuss the current HRD assays in the management of HGSOC. Overall, this Review manuscript is well organized and written. It will benefit both basic researchers and clinicians.
Some trivia suggestions.
- The authors might also want to consider the resistance aspects of PARPi in the treatment of OC patients.
- I do not see a necessity of Figure 1 in this Review. The authors might need to explain why they add this figure to the manuscript. Moreover, the lower panel of Figure 1 lists some factors but not all the actual essential ones in the specific DNA repair pathway. The authors should carefully check this panel (including some errors of the gene name) if they want to keep this figure.
- Line 271, “MMJE” should be “MMEJ”
Author Response
This is a part of two related reviews. In this review, the authors focus on the molecular and technical aspects of homologous recombination deficiency (HRD) in the diagnosis and treatment of patients with High-grade serous ovarian cancer (HGSOC). They summarize the causes and consequences of HRD in ovarian cancers. They also discuss the current HRD assays in the management of HGSOC. Overall, this Review manuscript is well organized and written. It will benefit both basic researchers and clinicians.
The authors kindly thank the reviewer for supporting comments and we will give answers through a point by point assay.
Some trivia suggestions.
- The authors might also want to consider the resistance aspects of PARPi in the treatment of OC patients.
The authors thank the reviewer for the interesting point regarding this highly relevant question. We actually decided to develop the “PARPi resistance” in the part2 of this review, as it is more clinically related. Furthermore, we put in the references already existing reviews that assessed comprehensively this question for readers interested by this focus.
- I do not see a necessity of Figure 1 in this Review. The authors might need to explain why they add this figure to the manuscript. Moreover, the lower panel of Figure 1 lists some factors but not all the actual essential ones in the specific DNA repair pathway. The authors should carefully check this panel (including some errors of the gene name) if they want to keep this figure.
The authors thank the reviewer for this relevant point. We do agree with your remark and removed the figure 1 of our review and simply replaced it with an introductory sentence :
“Briefly, single-strand breaks (SSB) are processed through base excision repair (BER) mechanism, mainly involving PARP proteins (of whom PARP1 is the most characterized). When facing double-strand break (DSB), cells will use different repair mechanisms: homologous recombination (HR), non-homologous end joining (NHEJ) and microhomology-mediated end joining (MMEJ). While the HR process leads to faithful DNA repair by using a homologous template, end resects are directly ligated by NHEJ, potentially leading to small insertions/deletions (indels). MMEJ also leads to specific indels (which are longer than the ones occurring during NHEJ).
- Line 271, “MMJE” should be “MMEJ”
The authors thank the reviewer for noticing this misprint
Reviewer 2 Report
The review is very interesting and very good structurate. I suggest few minor language and editing corrections.
Author Response
The authors kindly thank the reviewer for supporting comments, meanwhile we do not see the rest of the comments
Reviewer 3 Report
This manuscript provides a comprehensive review of homologous recombination deficiency (HRD) status in high-grade serous ovarian cancer (HGSOC) and discuss, in technical viewpoints, the pros and cons of the companion HRD assays for clinical management of HGSOC. The need and prospective in the refinement and new development of HRD assays for clinical use are also discussed. The content is suitable for the audience after revising the followings:
Section 2.2, add “in the HR pathway” to the end of the 1st sentence. This section still mentions about genetic alterations of BRCA1/2 but without explanations of the roles of BRCA1/2 in HGSOC development, treatment response, and patient outcome, which can be added.
Section 2.3, lines 154-159, without explaining the connection between long non-coding RNAs and HRD in term of BRCAness.
Fig 2, please define signature 3 in figure legend.
Line 222, “LAI” or “LST”?
Section 2.4.3, please give some real world examples to emphasize the applications of synthetic lethality and the clinical benefit of PARP inhibitors in ovarian cancer patients with HRD, which are prerequisite considerations for the discussion of section 3, HRD assays.
Table 1, letters are too small.
Please revise the sentence (line 313): “the mutation statuses of BRCA1 and BRCA2 are frequently evaluated 313 through multipanel testing, owing to the diminished cost of these kinds of tests and the 314 potently targetable non-BRCA alterations they can detect.”
Please give reference citations for lines 365-382, 496-535.
Author Response
[1]This manuscript provides a comprehensive review of homologous recombination deficiency (HRD) status in high-grade serous ovarian cancer (HGSOC) and discuss, in technical viewpoints, the pros and cons of the companion HRD assays for clinical management of HGSOC. The need and prospective in the refinement and new development of HRD assays for clinical use are also discussed. The content is suitable for the audience after revising the followings:
The authors kindly thank the reviewer for supporting comments and we will give answers through a point by point assay.
Section 2.2, add “in the HR pathway” to the end of the 1st sentence.
The authors kindly thank the reviewer for this comment, we added this end of sentence
This section still mentions about genetic alterations of BRCA1/2 but without explanations of the roles of BRCA1/2 in HGSOC development, treatment response, and patient outcome, which can be added.
We added the following sentnces : BRCA1/2 play key roles in genome integrity maintenance and their alteration are early event in EOC carcinogenetic process; indeed, loss of the first allele of BRCA1 (or BRCA2) is a facilitating event for TP53 loss, both of them leading to EOC development
Regarding the details of carcinogenesis related to BRCA loss, we do think that it is out of scope of our review and that it has been extensively developed elsewere. The clinical section is developed in the part 2 of our review.
Section 2.3, lines 154-159, without explaining the connection between long non-coding RNAs and HRD in term of BRCAness.
The authors kindly thank the reviewer for this comment. Actually, mechansims are not exactly deciphered regarding causes versus consequences when referring to lncRNAs, unless for in vitro and a few in vivo examples. We added 2 references in the review : one referring to recent review which developed extensively the known interactions between lnc/RNA/miRNA and HRD and another one showing HRD through PCAT-1 in prostate cancer
Fig 2, please define signature 3 in figure legend.
The authors kindly thank the reviewer for this comment, we added “Signature 3 refers to a specific base substitution pattern, which is the consequence of HRD.”
Line 222, “LAI” or “LST”?
The authors kindly thank the reviewer for this comment, it is a misprint that has been changed
Section 2.4.3, please give some real world examples to emphasize the applications of synthetic lethality and the clinical benefit of PARP inhibitors in ovarian cancer patients with HRD, which are prerequisite considerations for the discussion of section 3, HRD assays.
The authors kindly thank the reviewer for this comment, we added these sentences : [1]. In the context of HGSOC, seminal studies showed both in vivo and in vitro a specific SL occurring with PARPis in a BRCA deficient context [2,3]. Subsequently, this breakthrough class of agents started to emerge in randomized clinical trials with substantial improvements in patients with HRD HGSOC, leading to approval of three molecules to date: olaparib, niraparib and rucaparib. Clinical considerations will be developed in the part 2 of this review. Noteworthy, HRD HGSOC spontaneously tends to have improved progression-free survival (PFS) and OS, which is in part due to better treatment responses [4].
Regarding clinics, we developed in an extensive way the known data in the part 2
Table 1, letters are too small.
The authors kindly thank the reviewer for this comment, we changed the font
Please revise the sentence (line 313): “the mutation statuses of BRCA1 and BRCA2 are frequently evaluated 313 through multipanel testing, owing to the diminished cost of these kinds of tests and the 314 potently targetable non-BRCA alterations they can detect.”
The authors kindly thank the reviewer for this comment, we changed towards this sentence : Currently, the mutation statuses of BRCA1 and BRCA2 can be either evaluated through gene specific (i.e. BRCA1/2 targeted sequencing) or multipanel testing, the latter allowing to detect potently targetable non-BRCA alterations.
Please give reference citations for lines 365-382, 496-535.
The authors kindly thank the reviewer for this comment, actually all the data are given within the technical information sheets provided by MyriadGenetics and FoundationMedicine and that are already referenced in our review
Reviewer 4 Report
Excellent and very well-written review providing a deep analysis of the molecular mechanisms involved in the deficiency of homologous recombination (HR) in high-grade serious ovarian cancer (HGSOC) and their functional consequences. In addition to the well-known facts illustrating the high sensitivity of HR-defective (HRD) malignancies to the PARP inhibitors, the authors describe and explain novel mechanisms that might be considered as attractive targets for therapy of HRD tumors, including HGSOC. As expected, the current view about the assays used to examine HRD is described in detail and also includes the technical issues, concerns, and perspectives.
Author Response
The authors sincerely thank the reviewer for these encouraging comments.
Reviewer 5 Report
This review addresses HRD in ovarian cancer with emphasis on HRD assays and technical consideration. The review is well written and comprehensive. The reader will be updated with advantages and limitation of existing tests and provided insight toward needs in the field.
One minor error that would likely be found in editing; figure 2 legend has the word 'scares' I believe should be 'squares'.
Author Response
This review addresses HRD in ovarian cancer with emphasis on HRD assays and technical consideration. The review is well written and comprehensive. The reader will be updated with advantages and limitation of existing tests and provided insight toward needs in the field.
The authors sincerely thank the reviewer for these comments.
One minor error that would likely be found in editing; figure 2 legend has the word 'scares' I believe should be 'squares'.
Indeed, thanks for the remark, it is a misprint that has been changed